# ERC-SVD: Error-Controlled SVD for Large Language Model Compression

## Abstract

Large language models (LLMs) have demonstrated impressive capabilities in a wide range of downstream natural language processing tasks. Nevertheless, their considerable sizes and memory demands hinder practical deployment, underscoring the importance of developing efficient compression strategies. Singular value decomposition (SVD) decomposes a matrix into orthogonal components, enabling efficient low-rank approximation. This is particularly suitable for LLM compression, where weight matrices often exhibit significant redundancy. However, current SVD-based methods neglect the residual matrix from truncation, resulting in significant truncation loss. Additionally, compressing all layers of the model results in severe error propagation. To overcome these limitations, we propose **ERC-SVD**, a new post-training SVD-based LLM compression method from an error-controlled perspective. Specifically, we leverage the residual matrix generated during the truncation process to reduce truncation loss. Moreover, under a fixed overall compression ratio, we selectively compress the last few layers of the model, which mitigates error propagation and improves compressed model performance. Comprehensive evaluations on diverse LLM families and multiple benchmark datasets indicate that ERC-SVD consistently achieves superior performance over existing counterpart methods, demonstrating its practical effectiveness.

## 1 Introduction

Large language models (LLMs) have emerged as powerful tools, delivering state-of-the-art performance across a wide range of tasks such as text generation, translation, and reasoning. The scaling law (Kaplan et al., 2020) has driven a trend toward increasingly large models, exemplified by models such as GPT (Brown et al., 2020), PaLM (Chowdhery et al., 2023), LLaMA (Touvron et al., 2023a), Deepseek (Liu et al., 2024a), and Qwen (Yang et al., 2025), which often contain tens to hundreds of billions of parameters. Despite their powerful capabilities, the enormous scale of LLMs poses serious challenges for efficient deployment due to high computational demands (Sheng et al., 2023; Zhou et al., 2024; Wang et al., 2024). This resource burden not only limits deployment on edge devices and consumer-level hardware but also increases the cost and carbon footprint of serving LLMs in production (Strubell et al., 2020; Patterson et al., 2022).

As the scale of LLMs continues to grow, compression techniques, including weight quantization (Frantar et al., 2022; Lin et al., 2024; Huang et al., 2024; Li et al., 2025b), network pruning (Ma et al., 2023; Frantar & Alistarh, 2023; Sun et al., 2024; Gao et al., 2024), knowledge distillation (Gu et al., 2024; Yang et al., 2024; Xu et al., 2024; Zhang et al., 2024), and low-rank decomposition (Hsu et al., 2022; Kaushal et al., 2023; Yuan et al., 2023; Wang et al., 2025; Li et al., 2025a), have become increasingly important for the practical deployment of LLMs in resource-constrained environments. The syntactic and semantic correlations acquired during training often induce redundancy in LLM weight matrices, giving rise to a low-rank structure (Saha et al., 2024). As a result, singular value decomposition (SVD) provides a principled and effective approach for compressing these matrices with minimal loss. In particular, post-training approaches are gaining traction, as they can significantly reduce memory and compute requirements without the need for expensive retraining, making them especially suitable for scaling up foundation models. Recent studies on post-training SVD-based LLM compression, including ASVD (Yuan et al., 2023), SVD-LLM (Wang et al., 2025), and AdaSVD (Li et al., 2025a), have made significant progress in reducing model size while preserving performance, demonstrating the effectiveness of low-rank approximation techniques in compressing LLMs. More precisely,

while both ASVD and SVD-LLM apply weight matrix scaling before SVD truncation, SVD-LLM outperforms ASVD by leveraging a data whitening technique that enables a direct mapping between singular values and truncation loss. As a further advancement, AdaSVD compensates for truncation loss by iteratively updating singular matrices. However, existing methods suffer from two major limitations. First, existing methods ignore the importance of the residual matrix generated during SVD truncation, leading to significant truncation loss. Second, compressing all model layers often results in high layer-wise error and severe error propagation, as confirmed by our results in Figure 4.

In this work, we propose **ERC-SVD**, a new post-training SVD-based compression method for LLMs. Building upon the two key observations outlined above, ERC-SVD introduces two core technical innovations. ① **Residual compensation for SVD truncation:** The residual matrix produced during SVD truncation can be effectively utilized to reduce the truncation loss. Specifically, we perform SVD truncation in two stages: we first truncate the original weight matrix $W$ to obtain its intermediate low-rank approximation $W_{r_i}$. After that, we compute the residual matrix $R$ between $W$ and $W_{r_i}$. Second, we apply SVD truncation to $R$, yielding $R_{r_r}$. Finally, we construct the compressed weight matrix $\hat{W}_r = W_{r_i} + R_{r_r}$, where the total rank satisfies $r_i + r_r = r$. Detailed description is provided in Section 3.1. ② **Partial-layer compression for SVD:** LLMs consist of a sequence of consecutive layers, where the output of each layer serves as the input to the next. As a result, any error introduced in earlier layers can propagate and accumulate through subsequent layers, leading to severe degradation in performance. To mitigate this, we propose compressing only the last few layers under a fixed overall compression ratio while keeping the earlier layers intact. This strategy ensures that the earlier layers remain error-free, thereby reducing the impact of error propagation.

Our key contributions can be summarized as follows:

- We introduce **residual compensation for SVD truncation**, a theoretically grounded compensation strategy. By leveraging the residual matrix generated during the SVD truncation, our strategy significantly reduces the overall truncation loss.

- We propose **partial-layer compression for SVD**, which compresses only the last few layers of the model under a fixed overall compression ratio. This strategy effectively reduces layer-wise error and mitigates error propagation.

- Extensive experiments on multiple LLMs (LLaMA, OPT, Mistral, Vicuna, and Qwen) and benchmark datasets (both language modeling and zero-shot reasoning) demonstrate that ERC-SVD outperforms existing methods across a wide range of compression ratios.

## 2 RELATED WORK

### 2.1 TECHNIQUES FOR LARGE LANGUAGE MODEL COMPRESSION

The growing size of large language models (LLMs) has raised increasing concerns over their computational and memory demands. To address these challenges, a variety of model compression techniques have been proposed. Conventional approaches often require computationally expensive retraining, which is generally impractical due to the substantial computational cost associated with the massive size of LLMs. Consequently, recent efforts have shifted toward more resource-friendly post-training compression techniques (Frantar et al., 2022; Frantar & Alistarh, 2023; Zhu et al., 2024). Typically employed approaches include network pruning and weight quantization. And pruning techniques can be classified into unstructured and structured methods. Unstructured pruning removes individual weights based on importance scores. For example, SparseGPT (Frantar & Alistarh, 2023) performs one-shot pruning using second-order approximations without retraining. However, since unstructured pruning retains the original matrix shape, it offers limited inference acceleration and requires specialized hardware. In contrast, structured pruning eliminates entire blocks or channels, enabling compatibility with conventional hardware platforms. LLM-Pruner (Ma et al., 2023) groups dependent linear projections into coupled structures, assigns each group a loss-aware importance score, prunes the least important groups, and applies LoRA fine-tuning to restore performance. Additionally, ZipLM (Kurtic et al., 2023) prioritizes pruning components that yield the worst trade-off between latency and loss, but this often causes notable performance degradation. Quantization provides another mainstream solution. GPTQ (Frantar et al., 2022) gradually quantizes and updates each weight using the Hessian matrix to minimize the quantization error. AWQ (Lin et al., 2024) preserves important weight channels by selecting reparameterization coefficients via grid search.

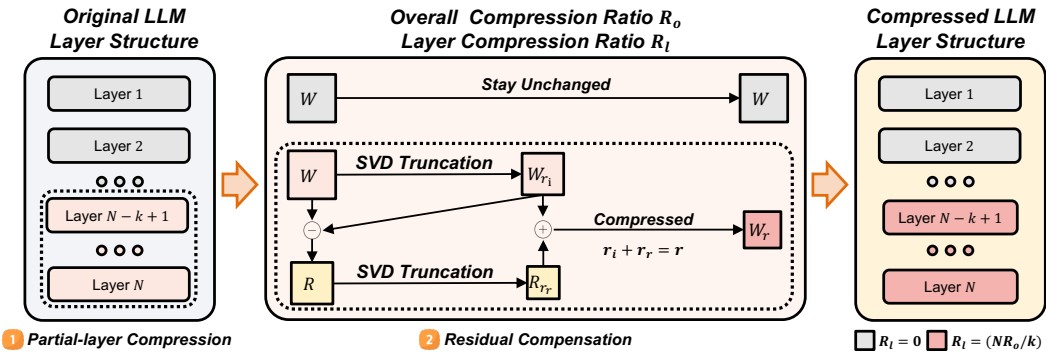

Figure 1: Framework of ERC-SVD. The last $k$ layers are selected through **partial-layer compression** and compressed using **residual compensation** with calibration data. The first $(N - k)$ layers remain intact, while the last $k$ layers are replaced by low-rank approximations.

Furthermore, BiLLM (Huang et al., 2024) and ARB-LLM (Li et al., 2025b) push quantization to the 1-bit level, while still delivering impressive accuracy across downstream tasks. However, these techniques still cause significant performance degradation, especially at low bit-widths, due to the lack of weight and activation adaptation.

## 2.2 SVD-BASED TECHNIQUES FOR COMPRESSING LLMS

Singular value decomposition (SVD) is a commonly employed technique for reducing matrix dimensionality by representing the original matrix as the product of two low-rank factor matrices. Recent research has demonstrated the potential of SVD-based LLM compression methods, yet comprehensive exploration remains limited. Hsu et al. (Hsu et al., 2022) propose FWSVD, incorporating Fisher information to weight the importance of parameters. However, it relies on computationally intensive training and was originally applied only to small language models (e.g., BERT (Devlin et al., 2019), ALBERT (Lan et al., 2020)). LoRD (Kaushal et al., 2023) first applies SVD to LLMs by grouping layers to improve efficiency. However, LoRD overlooks the importance of input activations. To address this limitation, Yuan et al. (Yuan et al., 2023) introduce ASVD, which mitigates the impact of activation outliers by reshaping the weight matrix based on activation distribution, thereby enhancing the precision of the low-rank decomposition. CALDERA (Saha et al., 2024) exploits the inherent low-rank structure of weight matrices by approximating them through a low-rank, low-precision decomposition. Additionally, SVD-LLM (Wang et al., 2025) establishes a direct mapping between singular values and truncation loss, which means that truncating the smallest singular values leads to minimal truncation loss. Dobi-SVD (Qinsi et al., 2025) provides theoretical and empirical evidence that truncating activations is more effective than truncating weights. More recently, AdaSVD (Li et al., 2025a) proposes an adaptive compensation method, iteratively updating matrices during truncation.

Despite these advances, all existing methods ignore the residual matrix produced during SVD truncation, which can significantly compensate for the SVD truncation loss. Moreover, compressing only the last few layers of the model under a fixed target compression ratio can provide better performance for compressed models. However, existing methods compress all layers, either apply a uniform compression ratio across all layers or assign variable ratios based on layer-wise importance, which often results in sub-optimal performance.

## 3 ERC-SVD

The framework of ERC-SVD is illustrated in Figure 1. We first perform SVD to compress the last few layers of the model, while ensuring that the overall compression ratio is satisfied, and compute the corresponding residual matrix. A second SVD is then applied to the residual matrix to obtain its low-rank approximation. The two truncated matrices are subsequently combined to construct the final compressed weight matrices. In Section 3.1, we first provide a detailed description of how **residual compensation** effectively works. Subsequently, in Section 3.2, we explain the benefits

---

**Algorithm 1** Pseudocode of ERC-SVD

---

**Input:** Original LLM: $M$, weight matrix: $\boldsymbol{W} \in \mathbb{R}^{m \times n}$, number of model layers: $N$, residual
    compensation rank $r_r$, step: $s$
**Output:** Compressed LLM $M'$ by ERC-SVD
 1: $CD \leftarrow$ Randomly collect calibration samples from the dataset
 2: $\text{Set}_{\boldsymbol{W}} \leftarrow M$, $\text{Set}_{\boldsymbol{W'}} \leftarrow \emptyset$                $\triangleright$ Initialize the sets of weight matrices
 3: $k, R_l \leftarrow$ PARTIAL-LAYER COMPRESSION$(M, N, R_o, s)$
 4: **for** Layer $i$ in original LLM $M$ **do**
 5:     **if** $i \in [1, N - k)$ **then**
 6:         $\text{Set}_{\boldsymbol{W'}}(i) \leftarrow \text{Set}_{\boldsymbol{W}}(i)$           $\triangleright$ Current weight matrices stay the same
 7:     **else**
 8:         $\boldsymbol{W}_i \leftarrow \text{Set}_{\boldsymbol{W}}(i)$            $\triangleright$ Initialize weight matrices to compress
 9:         $\text{Set}_{\boldsymbol{W'}}(i) \leftarrow$ RESIDUAL COMPENSATION$(M, \boldsymbol{W}_i, R_l, r_r)$
10:     **end if**
11:     $\text{Set}_{\boldsymbol{W'}} \leftarrow \text{Set}_{\boldsymbol{W'}}(i) \cup \text{Set}_{\boldsymbol{W'}}$         $\triangleright$ Append weight matrices after operation
12: **end for**
13: $M' \leftarrow$ UPDATE$(M, CD, \text{Set}_{\boldsymbol{W'}})$
14: **return** $M'$

---

of **partial-layer compression** in mitigating error propagation. The pseudocode of ERC-SVD is
shown in Algorithm 1, with the pseudocode for residual compensation and partial-layer compression
provided in Algorithm 2 and Algorithm 3, respectively.

### 3.1 RESIDUAL COMPENSATION FOR SVD TRUNCATION

**Preliminaries for SVD.** Typical SVD-based LLM compression methods apply SVD to the original
weight matrix $\boldsymbol{W} \in \mathbb{R}^{m \times n}$, and discard the smallest singular values to obtain a compressed low-rank
approximation $\boldsymbol{W}_r$:

$$\boldsymbol{W} = \boldsymbol{U}\boldsymbol{\Sigma}\boldsymbol{V}^T \approx \boldsymbol{U}_r\boldsymbol{\Sigma}_r\boldsymbol{V}_r^T = \boldsymbol{W}_r \, , r < \min(m, n), \tag{1}$$

where $\boldsymbol{U}_r \in \mathbb{R}^{m \times r}$ and $\boldsymbol{V}_r^T \in \mathbb{R}^{r \times n}$ are composed of the top-$r$ left and right singular vectors, respec-
tively, and $\boldsymbol{\Sigma}_r \in \mathbb{R}^{r \times r}$ is a diagonal matrix containing the corresponding singular values. Following
prior post-training SVD-based works (Yuan et al., 2023; Wang et al., 2025), the optimization objective
for SVD truncation in LLMs can be formulated as:

$$\hat{\boldsymbol{W}}_r = \arg\min_{\boldsymbol{W}_r} \left\| \boldsymbol{W}\boldsymbol{X} - \boldsymbol{W}_r\boldsymbol{X} \right\|_F \, , \tag{2}$$

$$\mathcal{L} = \left\| \boldsymbol{W}\boldsymbol{X} - \boldsymbol{W}_r\boldsymbol{X} \right\|_F = \left\| (\boldsymbol{W} - \boldsymbol{W}_r)\boldsymbol{X} \right\|_F \, , \tag{3}$$

where $\boldsymbol{X}$ denotes the activation of $\boldsymbol{W}$ given an input, and $\mathcal{L}$ is the truncation loss measured by
the Frobenius norm. Although previous works (Hsu et al., 2022; Yuan et al., 2023; Wang et al.,
2025) have made significant progress in minimizing $\mathcal{L}$, they consistently overlook the residual matrix
generated, despite its potential to compensate for the loss introduced by low-rank approximation.

Based on Equation 3, minimizing the truncation loss reduces to minimizing the discrepancy between
the original weight matrix $\boldsymbol{W}$ and its low-rank approximation $\boldsymbol{W}_r$. Accordingly, we reformulate the
optimization objective as:

$$\hat{\boldsymbol{W}}_r = \arg\min_{\boldsymbol{W}_r} \left\| \boldsymbol{W} - \boldsymbol{W}_r \right\|. \tag{4}$$

**Explanation for Residual Compensation.** Given the original weight matrix $\boldsymbol{W} \in \mathbb{R}^{m \times n}$ and the
layer compression ratio $R_l$, the compression ratio in this work denotes the fraction of parameters
removed. We define the effective scale of the matrix as:

$$\alpha = (m \cdot n)/(m + n), \tag{5}$$

where $m$ and $n$ represent the input and output dimensions of the matrix, hence, the target rank for
each layer is $r = (1 - R_l) \cdot \alpha$. We introduce a residual compensation factor $\beta$, a hyperparameter
fixed at $0.05$ in all experiments. The target rank $r$ is decomposed into two components: a residual
compensation rank $r_r$, defined as $r_r = \alpha \cdot \beta$, and an intermediate rank $r_i$, defined as $r_i = r - r_r$.

The entire compression process comprises two SVD truncation steps. First, we apply SVD to $\boldsymbol{W}$ and retain the top-$r_i$ singular values to obtain an intermediate low-rank approximation:

$$\boldsymbol{W}_{r_i} = \boldsymbol{U}_{r_i} \boldsymbol{\Sigma}_{r_i} \boldsymbol{V}_{r_i}^T. \tag{6}$$

The residual matrix is computed as the difference between the original matrix $\boldsymbol{W}$ and $\boldsymbol{W}_{r_i}$:

$$\boldsymbol{R} = \boldsymbol{W} - \boldsymbol{W}_{r_i}. \tag{7}$$

We then perform a second SVD on $\boldsymbol{R}$ and retain its top-$r_r$ singular values: $\boldsymbol{R}_{r_r} = \boldsymbol{U}_{r_r} \boldsymbol{\Sigma}_{r_r} \boldsymbol{V}_{r_r}^T$. Finally, we construct the compressed weight matrix by combining the two approximations:

$$\hat{\boldsymbol{W}}_r = \boldsymbol{W}_{r_i} + \boldsymbol{R}_{r_r} = \boldsymbol{U}_r \boldsymbol{\Sigma}_r \boldsymbol{V}_r^T = \hat{\boldsymbol{U}}_r \hat{\boldsymbol{V}}_r. \tag{8}$$

## 3.2 PARTIAL-LAYER COMPRESSION FOR SVD

Previous works (Hsu et al., 2022; Yuan et al., 2023; Wang et al., 2025; Li et al., 2025a) compress all model layers even if they assign layer-specific ratios based on their relative importance, which often leads to a high layer-wise error, resulting in noticeable degradation in the performance of compressed models. We compare the layer-wise error of LLMs across four families with different layer selection strategies; the results are shown in Figure 3. There is a significant error propagation across multiple LLM families, where the error progressively accumulates layer by layer during the forward pass. This phenomenon is particularly pronounced when compressing the first 8 layers. To satisfy the overall compression ratio, a higher layer compression ratio (80% in this case) must be applied to these layers. Compressing the model at such an early stage introduces substantial approximation errors, which are then propagated through the remaining layers. Although the later layers are left uncompressed, the forward pass carries the accumulated error through the network. Consequently, in all evaluated LLM families, these early compressed layers exhibit the highest layer-wise error across model layers. In contrast, when only compressing the last few layers, the earlier layers remain untouched with zero error. Although the compression ratio for selected layers is relatively high, leading to slightly faster error accumulation, the overall error remains significantly lower than that of compressing all layers.

Interestingly, we observe that although the final-layer error converges to a narrow range regardless of how many of the last layers are compressed, the differences within this range still exhibit a strong influence on model performance. As illustrated in the top figure of Figure 4, there is a strong correlation between the final-layer error and the average zero-shot accuracy. Motivated by this observation, we select the number of last layers to compress by minimizing the final-layer error, and the layer compression ratio $R_l$ satisfies:

$$R_l = (N \cdot R_o)/k, k \in \{1, 2, \ldots, N-1 \mid R_l < 1\}, \tag{9}$$

where $N$ is the number of model layers, $R_o$ represents the overall compression ratio. As shown in the bottom figure of Figure 4, compared to SVD-LLM (Wang et al., 2025), our strategy significantly reduces layer-wise error across all model layers and mitigates error propagation.

# 4 EXPERIMENTS

## 4.1 SETUPS

**Baselines.** We compare our method against four baselines without re-training: conventional SVD, ASVD (Yuan et al., 2023), SVD-LLM (Wang et al., 2025), and AdaSVD (Li et al., 2025a).

**Models and Datasets.** We evaluate ERC-SVD on ten models spanning five LLM families: LLaMA-7B (Touvron et al., 2023a), LLaMA-13B (Touvron et al., 2023a), LLaMA-30B (Touvron et al., 2023a), LLaMA-2-7B (Touvron et al., 2023b), LLaMA-2-13B (Touvron et al., 2023b), LLaMA-3-8B (Grattafiori et al., 2024), OPT-6.7B (Zhang et al., 2022), OPT-13B (Zhang et al., 2022), OPT-30B (Zhang et al., 2022), Mistral-7B (Jiang et al., 2023), Vicuna-7B (Chiang et al., 2023), Qwen-3-8B Yang et al. (2025). For language modeling, we use three benchmark datasets: WikiText-2 (Merity et al., 2017), PTB (Marcus et al., 1993), and C4 (Raffel et al., 2020). For zero-shot reasoning and understanding, we evaluate on seven tasks within the *LM-Evaluation-Harness* framework[1]:

---

[1] https://github.com/EleutherAI/lm-evaluation-harness

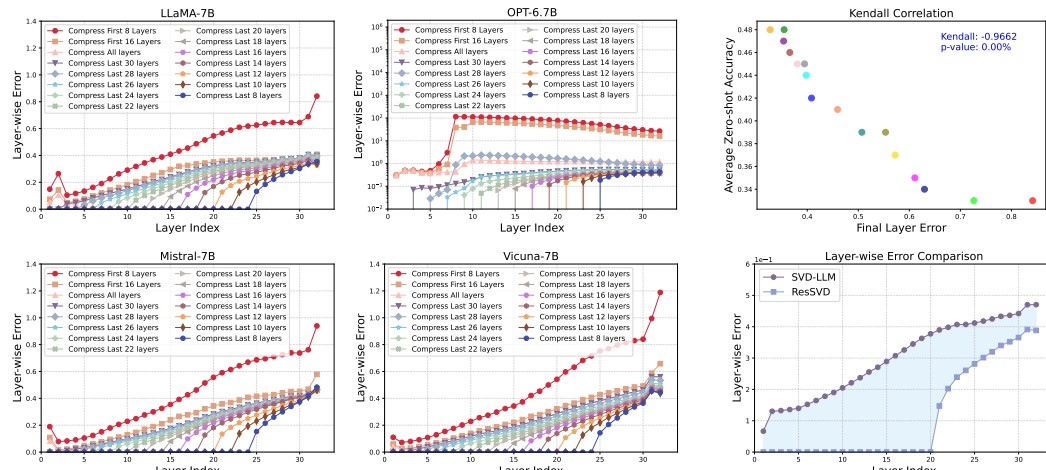

Figure 3: Layer-wise error comparison between original model and multiple LLM structures compressed by ERC-SVD with different layer selection strategies on WikiText-2. The overall compression ratio is 20%, and different layer selection strategies strictly adhere to the overall compression ratio constraint.

Figure 4: (Top) Kendall correlation. (Bottom) Layer-wise error comparison between ERC-SVD and SVD-LLM under 20% compression ratio.

OpenbookQA (Mihaylov et al., 2018), WinoGrande (Sakaguchi et al., 2021), HellaSwag (Zellers et al., 2019), PIQA (Bisk et al., 2020), MathQA (Amini et al., 2019), ARC_e and ARC_c (Clark et al., 2023). Moreover, to demonstrate the generality and effectiveness of ERC-SVD, we further extend our evaluation to vision-language models (VLMs), LLaVA (Liu et al., 2024b).

**Implementation Details.** To facilitate a fair comparison, we follow the protocols of ASVD, SVD-LLM, and AdaSVD, randomly selecting 256 calibration samples with a sequence length of 2048 from WikiText-2. We then apply data whitening prior to performing SVD truncation. All results are reproduced by re-running their respective open-source codebases, except for AdaSVD, whose results are directly taken from the original paper due to the lack of released code. All methods are all implemented with PyTorch[2] and Transformers[3] on NVIDIA A100 GPUs.

## 4.2 RESULTS

We evaluate the overall performance of ERC-SVD across four dimensions: ① **Effectiveness under different compression ratios** (ranging from 20% to 60%), ② **Generalizability across diverse LLM families**, ③ **Scalability to larger-scale models**, and ④ **Performance on VLMs**. In addition, qualitative results, such as generated contents and case studies of VLMs, are provided in Table 15 and Appendix A.7, respectively, providing a more intuitive comparison.

**Performance under different compression ratios.** We evaluate the performance of LLaMA-7B (Touvron et al., 2023a) and LLaMA-2-7B (Touvron et al., 2023b) compressed by ERC-SVD, conventional SVD, and existing post-training baselines under compression ratios ranging from 20% to 60% across ten benchmark datasets. The results for LLaMA-2-7B are presented in Table 1, while those for LLaMA-7B are provided in Table 9. Across three language modeling datasets, WikiText-2 (Merity et al., 2017), PTB (Marcus et al., 1993), and C4 (Raffel et al., 2020), ERC-SVD consistently outperforms all baselines across most evaluated compression ratios, with only slight suboptimalities observed in specific cases. In particular, on PTB and C4, the improvements are more pronounced, suggesting that ERC-SVD exhibits stronger generalization capability. More importantly, we can find significant improvements on PTB compared to the existing best-performing baseline SVD-LLM (Wang et al., 2025), with the largest improvement being 75% and the smallest improvement being 52%. In addition, on seven common sense reasoning datasets, ERC-SVD surpasses the existing

---

[2] https://github.com/pytorch/pytorch and HuggingFace
[3] https://github.com/huggingface/transformers

Table 1: Overall performance of LLaMA-2-7B (Touvron et al., 2023b) compressed by ERC-SVD and baselines under 20% to 60% compression ratios ("RATIO"), including performance on three language modeling datasets (measured by perplexity (↓)) and zero-shot performance on seven common sense reasoning datasets (measured by individual and average accuracy (↑)). The best results are marked in **bold**. Blue and green arrows within parentheses highlight the relative improvement over the existing second-best baseline. NaN denotes evaluation failure due to numerical instability. ∗ refers to results derived from the original paper. - means that results are not available.

| RATIO | METHOD | WikiText-2↓ | PTB↓ | C4↓ | Openb.↑ | ARC_e↑ | WinoG.↑ | HellaS.↑ | ARC_c↑ | PIQA↑ | MathQA↑ | Average↑ |
|---|---|---|---|---|---|---|---|---|---|---|---|---|
| - | Original | 5.47 | 26.84 | 7.28 | 0.31 | 0.69 | 0.67 | 0.56 | 0.40 | 0.78 | 0.27 | 0.53 |
| 20% | SVD | 18208.79 | 59320.78 | 27131.56 | 0.14 | 0.26 | 0.49 | 0.25 | 0.22 | 0.52 | 0.20 | 0.30 |
| | ASVD | 9.56 | 120.74 | **12.85** | 0.25 | 0.59 | 0.61 | 0.46 | 0.32 | 0.72 | 0.24 | 0.45 |
| | SVD-LLM | 8.37 | 139.68 | 20.18 | 0.23 | 0.50 | 0.59 | 0.39 | 0.26 | 0.65 | 0.23 | 0.41 |
| | **ERC-SVD** | **7.63** (↓9%) | **45.37** (↓62%) | 14.73 (-) | **0.28** | **0.61** | **0.65** | **0.50** | **0.35** | **0.73** | 0.26 | **0.48** (↑7%) |
| 30% | SVD | 30373.39 | 48930.94 | 36905.54 | 0.12 | 0.25 | 0.49 | 0.25 | 0.22 | 0.51 | 0.21 | 0.29 |
| | ASVD | 984.03 | NaN | NaN | 0.15 | 0.27 | 0.51 | 0.26 | 0.22 | 0.53 | 0.20 | 0.31 |
| | SVD-LLM | 10.66 | 292.90 | 34.96 | 0.21 | 0.42 | 0.55 | 0.34 | 0.22 | 0.60 | 0.23 | 0.37 |
| | **ERC-SVD** | **10.32** (↓3%) | **73.04** (↓75%) | **23.28** (↓33%) | **0.23** | **0.51** | **0.62** | **0.42** | **0.29** | **0.68** | **0.24** | **0.43** (↑16%) |
| 40% | SVD | 39524.00 | 68829.98 | 56518.95 | 0.13 | 0.26 | 0.50 | 0.25 | 0.21 | 0.52 | 0.18 | 0.29 |
| | ASVD | NaN | NaN | NaN | 0.15 | 0.25 | 0.50 | 0.26 | 0.22 | 0.52 | 0.18 | 0.30 |
| | SVD-LLM | 16.11 | 717.34 | 61.96 | 0.16 | 0.35 | 0.55 | 0.30 | 0.20 | 0.57 | 0.23 | 0.34 |
| | AdaSVD* | 14.76 | 304.62 | 56.98 | 0.19 | 0.41 | **0.58** | 0.32 | 0.23 | 0.58 | - | - |
| | **ERC-SVD** | **14.17** (↓4%) | **136.32** (↓55%) | **43.19** (↓24%) | **0.20** | **0.43** | 0.57 | **0.35** | **0.24** | **0.63** | 0.23 | **0.38** (↑12%) |
| 50% | SVD | 53405.48 | 39023.05 | 58547.82 | 0.15 | 0.25 | 0.48 | 0.25 | 0.22 | 0.53 | 0.18 | 0.29 |
| | ASVD | NaN | NaN | NaN | 0.13 | 0.26 | 0.50 | 0.25 | **0.23** | 0.50 | 0.20 | 0.30 |
| | SVD-LLM | 27.19 | 1775.52 | 129.71 | 0.14 | 0.30 | 0.50 | 0.28 | 0.20 | 0.54 | **0.23** | 0.31 |
| | AdaSVD* | 25.58 | 593.14 | 113.84 | **0.15** | 0.34 | 0.54 | 0.29 | 0.20 | 0.56 | - | - |
| | **ERC-SVD** | **24.26** (↓5%) | **286.24** (↓52%) | **100.34** (↓12%) | 0.14 | **0.35** | **0.55** | **0.31** | 0.22 | **0.59** | 0.22 | **0.34** (↑10%) |
| 60% | SVD | 65240.23 | 79002.21 | 70659.74 | 0.14 | 0.25 | 0.50 | 0.25 | 0.23 | 0.52 | 0.19 | 0.30 |
| | ASVD | NaN | 19581.17 | NaN | **0.15** | 0.25 | 0.50 | 0.25 | **0.23** | 0.52 | 0.12 | 0.29 |
| | SVD-LLM | **54.19** | 3442.74 | 263.02 | 0.14 | 0.26 | 0.50 | 0.27 | 0.20 | 0.53 | 0.21 | 0.30 |
| | AdaSVD* | 60.08 | 2137.28 | 294.26 | 0.12 | 0.27 | 0.50 | 0.27 | 0.20 | 0.53 | - | - |
| | **ERC-SVD** | 58.88 (-) | **991.48** (↓54%) | **255.70** (↓3%) | 0.13 | **0.29** | 0.52 | 0.28 | 0.21 | 0.55 | 0.22 | **0.31** (↑3%) |

**Performance on Multiple LLM families.** To evaluate the generalization ability of ERC-SVD, we apply it to four LLMs from different families, including OPT-6.7B, LLaMA-2-7B, Mistral-7B, and Vicuna-7B. As shown in Table 2, under 30% compression ratio, ERC-SVD consistently outperforms all baselines on three language modeling benchmarks across these diverse architectures. The most significant relative improvement reaches 75% on LLaMA-2-7B. Moreover, the best overall improvement is achieved on Mistral-7B, with perplexity reductions of 71% on WikiText-2, 45% on PTB, and 46% on C4. We reproduce ASVD and SVD-LLM using their public codebases. While ASVD fails in certain cases due to numerical instability (denoted as NaN in the table), ERC-SVD consistently maintains stable and reliable performance. More zero-shot accuracy results are provided in Table 13.

Table 2: Perplexity (↓) of different LLM structures under 30% compression ratio.

| MODEL | METHOD | WikiText-2↓ | PTB↓ | C4↓ |
|---|---|---|---|---|
| OPT-6.7B | SVD | 116067.28 | 86760.50 | 168165.89 |
| | ASVD | 26.67 | 71.36 | 44.51 |
| | SVD-LLM | 28.03 | 37.46 | 40.35 |
| | **ERC-SVD** | **17.10** (↓36%) | **27.24** (↓27%) | **38.40** (↓5%) |
| LLaMA-2-7B | SVD | 30373.39 | 48930.94 | 36905.54 |
| | ASVD | 984.03 | NaN | NaN |
| | SVD-LLM | 10.66 | 292.90 | 34.96 |
| | **ERC-SVD** | **10.32** (↓3%) | **73.04** (↓75%) | **23.28** (↓33%) |
| Mistral-7B | SVD | 59569.54 | 57830.63 | 78168.24 |
| | ASVD | 221.66 | 927.15 | 266.04 |
| | SVD-LLM | 48.94 | 193.22 | 56.55 |
| | **ERC-SVD** | **14.09** (↓71%) | **105.37** (↓45%) | **30.72** (↓46%) |
| Vicuna-7B | SVD | 24835.33 | 24510.90 | 29368.55 |
| | ASVD | 106.32 | NaN | NaN |
| | SVD-LLM | 12.42 | 104.27 | 39.55 |
| | **ERC-SVD** | **11.57** (↓7%) | **69.28** (↓34%) | **27.24** (↓31%) |

Table 3: Perplexity (↓) of larger-scale LLMs under 20% compression ratio on PTB (Marcus et al., 1993).

| METHOD | LLaMA-13B | LLaMA-30B | LLaMA-2-13B | OPT-13B | OPT-30B |
|---|---|---|---|---|---|
| SVD | 1878.04 | 555.55 | 5464.57 | 1552.55 | 250.49 |
| ASVD | 12.42 | 31.36 | 81.00 | 36.47 | 30.95 |
| SVD-LLM | 12.17 | 9.10 | 88.13 | 14.86 | 12.94 |
| **ERC-SVD** | **9.70** | **8.41** | **66.47** | **13.22** | **12.89** |

best baseline on the majority of datasets, with an average accuracy gain of up to 16%, further highlighting its robustness and overall effectiveness.

**Performance on larger-scale LLMs.** To examine the scalability of ERC-SVD, we evaluate its performance on larger-scale LLMs from two representative families: LLaMA and OPT (13B and 30B). As presented in Table 3, ERC-SVD consistently achieves superior performance over existing baselines under 20% compression ratio, demonstrating robust effectiveness across varying model scales. More zero-shot accuracy results on these larger-scale LLMs are provided in Table 11.

Table 4: Performance (measured by accuracy (↑)) of original LLaVA-1.5-7B, and its 20% compressed versions by SVD-LLM and ERC-SVD on VLM benchmarks. The best results are marked in **bold**.

| COMPRESSION RATIO | METHOD | POPE-random↑ | POPE-popular↑ | POPE-adversial↑ | TextVQA↑ | ScienceQA↑ |
|---|---|---|---|---|---|---|
| - | Original | 88.2 | 87.3 | 85.1 | 58.17 | 70.15 |
| 20% | SVD-LLM | 82.5 | 83.2 | 77.8 | 30.68 | 49.54 |
| | **ERC-SVD** | **90.2** (↑9%) | **87.7** (↑5%) | **83.1** (↑7%) | **50.86** (↑66%) | **69.54** (↑40%) |

Table 5: Ablation results of REC and PLC on LLaMA-2-7B under 30% and 40% compression ratios.

| RATIO | METHOD | REC | PLC | C4↓ | Openb.↑ | ARC_e↑ | WinoG.↑ | HellaS.↑ | ARC_c↑ | PIQA↑ | MathQA↑ | Average↑ |
|---|---|---|---|---|---|---|---|---|---|---|---|---|
| 30% | ASVD | - | - | NaN | 0.15 | 0.27 | 0.51 | 0.26 | 0.22 | 0.53 | 0.20 | 0.31 |
| | SVD-LLM | - | - | 34.96 | 0.21 | 0.42 | 0.55 | 0.34 | 0.22 | 0.60 | 0.23 | 0.37 |
| | **ERC-SVD** | ✓ | ✗ | 30.68 | 0.22 | 0.42 | 0.58 | 0.35 | 0.23 | 0.61 | 0.24 | 0.38 |
| | | ✗ | ✓ | 24.77 | 0.22 | 0.48 | 0.60 | 0.38 | 0.25 | 0.65 | 0.22 | 0.40 |
| | | ✓ | ✓ | **23.28** | 0.23 | 0.51 | 0.62 | 0.42 | 0.29 | 0.68 | 0.24 | **0.43** |
| 40% | ASVD | - | - | NaN | 0.15 | 0.25 | 0.50 | 0.26 | 0.22 | 0.52 | 0.18 | 0.30 |
| | SVD-LLM | - | - | 61.96 | 0.16 | 0.35 | 0.55 | 0.30 | 0.20 | 0.57 | 0.23 | 0.34 |
| | **ERC-SVD** | ✓ | ✗ | 54.19 | 0.17 | 0.37 | 0.54 | 0.32 | 0.21 | 0.57 | 0.23 | 0.34 |
| | | ✗ | ✓ | 45.13 | 0.18 | 0.40 | 0.55 | 0.33 | 0.24 | 0.61 | 0.21 | 0.36 |
| | | ✓ | ✓ | **43.19** | 0.20 | 0.43 | 0.57 | 0.35 | 0.24 | 0.63 | 0.23 | **0.38** |

**Performance on VLM.** We compare the performance of LLaVA-1.5-7B (Liu et al., 2024b) compressed using SVD-LLM (Wang et al., 2025) and ERC-SVD under 20% compression ratio. We report results on several benchmarks: POPE (Li et al., 2023), TextVQA (Singh et al., 2019), and ScienceQA (Lu et al., 2022). The results, shown in Table 4, indicate that ERC-SVD consistently outperforms SVD-LLM across all benchmarks. Notably, it achieves substantial relative improvements, 66% on TextVQA and 40% on ScienceQA. Moreover, on the POPE-random and POPE-popular subsets, the model compressed by ERC-SVD even surpasses the original LLaVA-1.5-7B.

## 4.3 ABLATION STUDY

We present several ablation studies to assess the robustness of ERC-SVD. ① **Effectiveness of residual compensation (REC) and partial-layer compression (PLC):** We assess the individual contributions of REC and PLC. ② **Impact of residual compensation factor:** We conduct experiments to examine how the choice of $\beta$ influences the performance. ③ **Impact of calibration data:** We analyze the effects of calibration dataset selection on compressed model performance.

**Effectiveness of residual compensation and partial-layer compression.** We present results on C4 (Raffel et al., 2020) and seven zero-shot reasoning and understanding tasks, as shown in Table 5. It can be observed that either applying REC or PLC alone has already improved performance. Combining them leads to further improvements, making the performance gap even more significant. Results of other compression ratios are presented in Table 12.

**Impact of residual compensation factor.** The residual compensation factor $\beta$ is a hyperparameter in this work. To examine its impact on model performance, we compress LLaMA-2-7B under 30% compression ratio while varying the value of $\beta$. The results in Table 6 show that the compressed models achieve comparable performance across different $\beta$ values, demonstrating the robustness of our method with respect to this hyperparameter.

Table 6: Perplexity (↓) and zero-shot individual and average accuracy (↑) of LLaMA-2-7B 30% compressed by ERC-SVD with different residual compensation factors ($\beta$).

| $\beta$ | WikiText-2↓ | Openb.↑ | ARC_e↑ | WinoG.↑ | HellaS.↑ | ARC_c↑ | PIQA↑ | MathQA↑ | Average↑ |
|---|---|---|---|---|---|---|---|---|---|
| 0.025 | 10.38 | 0.23 | 0.51 | 0.62 | 0.41 | 0.28 | 0.67 | 0.23 | 0.42 |
| 0.050 | 10.32 | 0.23 | 0.51 | 0.62 | 0.42 | 0.29 | 0.68 | 0.24 | **0.43** |
| 0.075 | **10.11** | 0.22 | 0.49 | 0.61 | 0.39 | 0.28 | 0.66 | 0.23 | 0.41 |
| 0.010 | 10.46 | 0.23 | 0.47 | 0.60 | 0.38 | 0.26 | 0.66 | 0.23 | 0.40 |

**Impact of calibration data.** We assess the selection of calibration data from two perspectives: both the number of calibration samples and the choice of calibration datasets. Figure 6 shows the perplexity on WikiText-2 (Merity et al., 2017) and average zero-shot accuracy under 30% compression

Table 7: Perplexity (↓) and zero-shot average accuracy (↑) of LLaMA-2-7B 20% compressed by ERC-SVD with different calibration datasets.

| CALI. DATA | WikiText-2↓ | PTB↓ | C4↓ | Avg.↑ |
|---|---|---|---|---|
| WikiText-2 | **7.63** | 45.37 | 14.73 | **0.48** |
| PTB | 9.71 | **29.43** | 13.29 | 0.46 |
| C4 | 9.44 | 37.67 | **11.49** | 0.47 |
| Mix | 10.13 | 36.53 | 13.13 | 0.45 |

Table 8: Perplexity (↓) of LLaMA-2-7B compressed by ERC-SVD and SVD-LLM, followed by quantization with GPTQ-8bit (Frantar et al., 2022). Blue arrows within parentheses highlight the relative improvement.

| RATIO | METHOD | QUANTIZATION | WikiText-2↓ | PTB↓ | C4↓ |
|---|---|---|---|---|---|
| - | Original | GPTQ-8bit | 5.47 | 26.79 | 7.28 |
| 30% | SVD-LLM | GPTQ-8bit | 10.67 | 336.30 | 34.93 |
| | **ERC-SVD** | GPTQ-8bit | **10.34** (↓3%) | **79.05** (↓76%) | **25.02** (↓28%) |

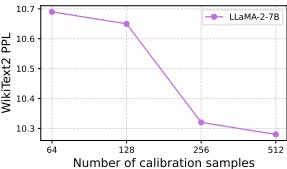
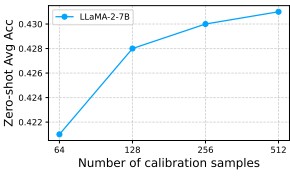
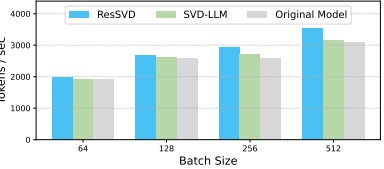

Figure 6: Impact of the number of calibration data samples on LLaMA-2-7B under 30% compression ratio. (Left) Perplexity (↓). (Right) Average accuracy (↑).

Figure 7: Throughput of LLaMA-7B and its 40% compressed versions. The sequence length is 32.

ratio. While performance slightly improves with more calibration samples, the gains are modest, indicating that ERC-SVD remains robust even with limited calibration data. Moreover, the best results are achieved when the calibration and evaluation datasets align, as shown in Table 7, suggesting that domain consistency enhances compression quality. We further evaluate using a mixed dataset, constructed by equally combining the three datasets, to examine its impact on model performance.

### 4.4 COMPATIBILITY WITH QUANTIZATION

SVD-based LLM compression methods and quantization are two orthogonal techniques. To demonstrate that our method can be integrated with quantization, we adopt GPTQ (Frantar et al., 2022) to quantize LLaMA-2-7B compressed by our method and SVD-LLM. As shown in Table 8, our method integrates seamlessly with quantization, achieving superior performance compared to SVD-LLM.

### 4.5 EFFICIENCY RESULTS

ERC-SVD not only preserves competitive model performance but also achieves substantial inference speedup on hardware. We evaluate the throughput of compressed models on an NVIDIA A100 GPU and present the results in Figure 7. Models compressed by ERC-SVD consistently deliver faster inference than the original model. Moreover, the speedup grows more pronounced as the batch size grows, indicating that ERC-SVD scales more efficiently under larger workloads. These findings highlight the practical effectiveness of ERC-SVD in enabling faster inference while maintaining accuracy, making it well-suited for deployment in resource-constrained environments. Additionally, Appendix A.6 provides further analysis on how ERC-SVD reduces computational complexity.

### 5 CONCLUSION

In this work, we propose ERC-SVD, a novel post-training SVD-based compression method for LLMs, formulated from an error-controlled perspective. ERC-SVD effectively leverages the residual matrix resulting from SVD truncation to reduce the truncation loss and enhance layer-wise reconstruction accuracy. Furthermore, it selectively compresses only the last few layers of the model under a fixed overall compression ratio, thereby significantly mitigating error propagation across the model. Extensive experiments across various LLM families and benchmark datasets demonstrate that ERC-SVD consistently outperforms existing SVD-based baselines under various settings. Results on VLM further demonstrate its effectiveness. Moreover, ERC-SVD integrates seamlessly with quantization, enabling further compression. These results highlight the effectiveness and generalizability of ERC-SVD in enabling efficient LLM deployment.

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

# A  APPENDIX

Here, we provide further details that are not discussed in the main paper and include extra experimental results. The appendix is structured as follows:

## A.1  PSEUDOCODE

Algorithm 2 and Algorithm 3 present the pseudocode for **residual compensation** and **partial-layer compression**, respectively. Algorithm 3 identifies the optimal number of last layers $k$ to compress, along with their corresponding layer compression ratio $R_l$. Specifically, for a model with $N$ layers, we iterate over candidate values of $k'$ using a step size $s$ and compute the associated $R'_l$ for each. For each candidate configuration, we invoke Algorithm 2 to perform the compression, after which we compute the final-layer error relative to the original model. The configuration yielding the lowest final-layer error is selected as the optimal compression setting.

---

**Algorithm 2** Pseudocode of Residual Compensation

---

**Input:** Original LLM: $M$, weight matrix: $\boldsymbol{W}_i \in \mathbb{R}^{m \times n}$, layer compression ratio: $R_l$, residual
    compensation rank: $r_r$
**Output:** Compressed weight matrix set $\text{Set}_{\boldsymbol{W'}}(i)$ of layer $i$
 1: $CD \leftarrow$ Randomly collect calibration samples from the dataset
 2: $\text{Set}_{\boldsymbol{S}} \leftarrow \text{WHITENING}(M, CD)$, $\text{Set}_{\boldsymbol{W'}}(i) \leftarrow \emptyset$        ▷ Initialize sets of weight matrices
 3: $r = (1 - R_l)(m \cdot n)/(m + n)$, $r_i = r - r_r$        ▷ Calculate the intermediate rank
 4: $\boldsymbol{S}_i \leftarrow \text{Set}_{\boldsymbol{S}}(i)$        ▷ Extract whitening matrices of current weight matrices
 5: $\boldsymbol{U}_{i,r_i}, \boldsymbol{\Sigma}_{i,r_i}, \boldsymbol{V}_{i,r_i}^T \leftarrow \boldsymbol{W}_{i,r_i} \leftarrow \text{SVD\_TRUNC}(\boldsymbol{W}_i\boldsymbol{S}_i)$    ▷ SVD truncation on weight matrices
 6: $\boldsymbol{R}_i \leftarrow \text{CAL\_RES}(\boldsymbol{W}_i, \boldsymbol{W}_{i,r_i})$        ▷ Calculate residual matrices
 7: $\boldsymbol{U}_{i,r_r}, \boldsymbol{\Sigma}_{i,r_r}, \boldsymbol{V}_{i,r_r}^T \leftarrow \boldsymbol{R}_{i,r_r} \leftarrow \text{SVD\_TRUNC}(\boldsymbol{R}_i)$    ▷ SVD truncation on residual matrices
 8: $\hat{\boldsymbol{U}}_{i,r_r} \leftarrow \text{MUL}(\boldsymbol{U}_{i,r_r}, \sqrt{\boldsymbol{\Sigma}_{i,r_r}})$, $\hat{\boldsymbol{V}}_{i,r_r} \leftarrow \text{MUL}(\sqrt{\boldsymbol{\Sigma}_{i,r_r}}, \boldsymbol{V}_{i,r_r}^T)$    ▷ Absorb singular values
 9: $\hat{\boldsymbol{U}}_{i,r_i} \leftarrow \text{MUL}(\boldsymbol{U}_{i,r_i}, \sqrt{\boldsymbol{\Sigma}_{i,r_i}})$, $\hat{\boldsymbol{V}}_{i,r_i} \leftarrow \text{MUL}(\sqrt{\boldsymbol{\Sigma}_{i,r_i}}, \boldsymbol{V}_{i,r_i}^T)$
10: $\hat{\boldsymbol{U}}_{i,r} \leftarrow \text{COMB}(\hat{\boldsymbol{U}}_{i,r_r}, \hat{\boldsymbol{U}}_{i,r_i})$, $\hat{\boldsymbol{V}}_{i,r} \leftarrow \text{COMB}(\hat{\boldsymbol{V}}_{i,r_r}, \hat{\boldsymbol{V}}_{i,r_i})$    ▷ Combine weight matrices
    $\text{Set}_{\boldsymbol{W'}}(i) \leftarrow (\hat{\boldsymbol{U}}_{i,r}, \hat{\boldsymbol{V}}_{i,r}) \cup \text{Set}_{\boldsymbol{W'}}(i)$        ▷ Append decomposed weight matrices
11: **return** $\text{Set}_{\boldsymbol{W'}}(i)$

---

## A.2  OVERALL PERFORMANCE OF LLaMA-7B

The evaluation results of LLaMA-7B (Touvron et al., 2023a) are reported in Table 9. On the three language modeling datasets, WikiText-2 (Merity et al., 2017), PTB (Marcus et al., 1993),

---

**Algorithm 3** Pseudocode of Partial-layer Compression

---

**Input:** Original LLM: $M$, number of model layers: $N$, overall compression ratio: $R_o$, step: $s$
**Output:** Number of layers to compress: $k$, layer compression ratio: $R_l$
1: $CD \leftarrow$ Randomly collect calibration samples from the dataset
2: $\text{Set}_{\boldsymbol{W}} \leftarrow M$                          ▷ Extract the set of weight matrices in $M$
3: $\text{Set}_{k'} \leftarrow \{\, k' \mid k' = s, 2s, \ldots, N-s \,\}$         ▷ Obtain candidate compressed layer numbers
4: $Lowest\_Err \leftarrow +\infty$
5: **for** each $k'$ in $\text{Set}_{k'}$ **do**
6:     $R'_l \leftarrow (N \cdot R_o)/k'$                     ▷ Calculate corresponding layer compression ratio
7:     **if** $R'_l \geq 0$ **then**
8:        $M_{TMP} \leftarrow \text{DEEP\_COPY}(M)$
9:        **for** $i \in [N - k' + 1, N]$ **do**
10:           $\mathbf{W}_i \leftarrow \text{Set}_{\boldsymbol{W}}(i)$                 ▷ Initialize weight matrices to compress
11:           $\text{Set}_{\boldsymbol{W}'}(i) \leftarrow \text{RESIDUAL COMPENSATION}(M, \mathbf{W}_i, R'_l, r_r)$
12:           Replace weights of layer $i$ in $M_{TMP}$ with $\text{Set}_{\boldsymbol{W}'}(i)$
13:        **end for**
14:        $Err \leftarrow \text{CAL\_ERROR}(M, M_{TMP}, CD, Layer(N))$ ▷ Calculate the last layer output error
15:        **if** $Err < Lowest\_Err$ **then**
16:           $Lowest\_Err \leftarrow Err$                  ▷ Search for the lowest layer-wise error
17:           $k \leftarrow k'$
18:           $R_l \leftarrow R'_l$
19:        **end if**
20:     **end if**
21: **end for**
22: **return** $k, R_l$

---

and C4 (Raffel et al., 2020), ERC-SVD consistently outperforms all baselines across evaluated compression ratios. In particular, on PTB and C4, the improvements are more pronounced, suggesting that ERC-SVD exhibits stronger generalization capability. More importantly, under a relatively high compression ratio (e.g., 50%), ERC-SVD still achieves substantial perplexity reductions compared to the existing best-performing baseline SVD-LLM (Wang et al., 2025): 7% on WikiText-2, 14% on PTB, and 41% on C4. This demonstrates that ERC-SVD maintains an obvious performance advantage even under aggressive compression. In addition, on seven common sense reasoning datasets, ERC-SVD surpasses the existing best baseline on the majority of datasets, with an average accuracy gain of up to 9%, and a minimum improvement of 3%.

### A.3 ADDITIONAL RESULTS

In this section, we present additional results from three perspectives: Section A.3.1 reports zero-shot accuracy on larger-scale LLMs, Section A.3.2 reports zero-shot accuracy across multiple LLM families, and Section A.3.3 presents ablation results under compression ratios 50% and 60%.

#### A.3.1 RESULTS ON LARGER-SCALE LLMS

In Table 11, we present zero-shot accuracy results for LLaMA-13B (Touvron et al., 2023a), LLaMA-30B (Touvron et al., 2023a), LLaMA-2-13B (Touvron et al., 2023b), OPT-13B (Zhang et al., 2022), and OPT-30B (Zhang et al., 2022). Across these evaluation datasets, ERC-SVD consistently outperforms SVD-LLM, with only one minor exception.

#### A.3.2 RESULTS ON MULTIPLE LLM FAMILIES

We also evaluate the zero-shot accuracy of OPT-6.7B (Zhang et al., 2022), Mistral-7B (Jiang et al., 2023), and Vicuna-7B (Chiang et al., 2023). Results are shown in Table 13, and our method consistently outperforms SVD-LLM across these diverse LLM architectures. In addition, we conduct experiments on two recent LLMs: LLaMA-3-8B (Grattafiori et al., 2024) and Qwen-3-8B (Yang et al., 2025), with perplexity and zero-shot accuracy results reported in Table 10. ERC-SVD also demonstrates consistent improvements over SVD-LLM across these evaluation tasks.

Table 9: Overall performance of LLaMA-7B compressed by ERC-SVD and baselines under 20% to 60% compression ratios ("RATIO"), including performance on three language modeling datasets (measured by perplexity ($\downarrow$)) and zero-shot performance on seven common sense reasoning datasets (measured by individual and average accuracy ($\uparrow$)). The best results are marked in **bold**. Blue arrows within parentheses highlight the relative improvement over the second-best method.

| RATIO | METHOD | WikiText-2↓ | PTB↓ | C4↓ | Openb.↑ | ARC_e↑ | WinoG.↑ | HellaS.↑ | ARC_c↑ | PIQA↑ | MathQA↑ | Average↑ |
|---|---|---|---|---|---|---|---|---|---|---|---|---|
| - | Original | 5.67 | 8.80 | 7.63 | 0.28 | 0.67 | 0.67 | 0.56 | 0.38 | 0.78 | 0.27 | 0.52 |
| 20% | SVD | 20082.86 | 20338.96 | 18784.20 | 0.14 | 0.27 | 0.51 | 0.26 | 0.21 | 0.53 | 0.21 | 0.30 |
| | ASVD | 9.27 | 15.09 | 13.68 | 0.25 | 0.53 | 0.60 | 0.41 | 0.27 | 0.68 | 0.25 | 0.43 |
| | SVD-LLM | 7.89 | 16.54 | 15.92 | 0.23 | 0.56 | 0.62 | 0.42 | 0.29 | 0.69 | 0.23 | 0.43 |
| | **ERC-SVD** | **7.47** (↓5%) | **12.27** (↓19%) | **12.22** (↓11%) | **0.25** | **0.59** | **0.67** | **0.47** | **0.34** | **0.71** | **0.25** | **0.47** (↑9%) |
| 30% | SVD | 13155.97 | 17354.46 | 21012.91 | 0.13 | 0.26 | 0.51 | 0.25 | 0.21 | 0.54 | 0.21 | 0.30 |
| | ASVD | 222.98 | 586.79 | 148.79 | 0.15 | 0.32 | 0.53 | 0.30 | 0.21 | 0.59 | 0.21 | 0.33 |
| | SVD-LLM | 9.52 | 28.97 | 26.38 | 0.20 | 0.49 | 0.59 | 0.37 | 0.27 | 0.65 | 0.22 | 0.40 |
| | **ERC-SVD** | **9.52** (-) | **20.31** (↓30%) | **18.29** (↓31%) | **0.23** | **0.54** | **0.63** | **0.41** | **0.30** | **0.67** | **0.23** | **0.43** (↑8%) |
| 40% | SVD | 52326.99 | 59859.41 | 47643.04 | 0.15 | 0.25 | 0.51 | 0.25 | 0.21 | 0.52 | 0.20 | 0.30 |
| | ASVD | 5262.11 | 8806.33 | 6522.61 | 0.14 | 0.26 | 0.49 | 0.26 | 0.22 | 0.53 | 0.20 | 0.30 |
| | SVD-LLM | 13.83 | 57.07 | 48.47 | 0.19 | 0.41 | 0.58 | 0.32 | 0.22 | 0.59 | 0.22 | 0.36 |
| | **ERC-SVD** | **12.92** (↓7%) | **46.93** (↓18%) | **30.51** (↓37%) | **0.19** | **0.44** | **0.60** | **0.35** | **0.27** | **0.62** | **0.24** | **0.39** (↑8%) |
| 50% | SVD | 130388.72 | 86721.38 | 79853.46 | 0.16 | 0.26 | 0.49 | 0.25 | 0.22 | 0.52 | 0.18 | 0.30 |
| | ASVD | 62726.86 | 117959.06 | 77773.84 | 0.13 | 0.25 | 0.48 | 0.25 | 0.22 | 0.53 | 0.21 | 0.30 |
| | SVD-LLM | 24.05 | 150.58 | 141.87 | 0.16 | 0.34 | 0.55 | 0.29 | 0.21 | 0.56 | 0.22 | 0.33 |
| | **ERC-SVD** | **22.41** (↓7%) | **128.80** (↓14%) | **83.80** (↓41%) | **0.16** | **0.36** | **0.58** | **0.32** | **0.23** | **0.59** | **0.22** | **0.35** (↑6%) |
| 60% | SVD | 52326.99 | 59859.41 | 47643.04 | 0.15 | 0.25 | 0.51 | 0.25 | 0.21 | 0.52 | 0.20 | 0.30 |
| | ASVD | 16221.43 | 20119.36 | 16561.39 | 0.13 | 0.26 | 0.50 | 0.25 | **0.23** | 0.53 | 0.18 | 0.30 |
| | SVD-LLM | 53.20 | 378.19 | 310.17 | 0.12 | 0.29 | 0.52 | 0.28 | 0.20 | 0.55 | **0.22** | 0.31 |
| | **ERC-SVD** | **48.67** (↓9%) | **323.74** (↓14%) | **260.06** (↓16%) | 0.14 | **0.30** | **0.53** | **0.28** | 0.21 | **0.55** | 0.21 | **0.32** (↑3%) |

Table 10: Performance of LLaMA-3-8B and Qwen-3-8B under 20% compression ratio.

| MODEL | METHOD | WikiText-2↓ | PTB↓ | C4↓ | Openb.↑ | ARC_e↑ | WinoG.↑ | HellaS.↑ | ARC_c↑ | PIQA↑ | MathQA↑ | Average↑ |
|---|---|---|---|---|---|---|---|---|---|---|---|---|
| LLaMA-3-8B | Original | 6.13 | 9.91 | 9.46 | 0.35 | 0.80 | 0.72 | 0.60 | 0.50 | 0.79 | 0.40 | 0.59 |
| | SVD-LLM | 47.17 | 51.04 | 81.96 | 0.16 | 0.49 | 0.53 | 0.32 | 0.22 | 0.64 | 0.24 | 0.37 |
| | **ERC-SVD** | **33.02** | **45.18** | **52.01** | **0.20** | **0.55** | **0.65** | **0.40** | **0.30** | **0.67** | **0.28** | **0.44** |
| Qwen-3-8B | Original | 9.71 | 15.43 | 15.52 | 0.31 | 0.83 | 0.68 | 0.57 | 0.56 | 0.77 | 0.49 | 0.60 |
| | SVD-LLM | 37.52 | **40.73** | 47.25 | **0.20** | 0.50 | 0.55 | 0.35 | 0.24 | 0.64 | 0.22 | 0.39 |
| | **ERC-SVD** | **35.11** | 42.77 | **43.10** | 0.19 | **0.54** | **0.58** | **0.35** | **0.25** | **0.66** | **0.24** | **0.40** |

### A.3.3 ADDITIONAL ABLATION RESULTS

The ablation results for LLaMA-2-7B (Touvron et al., 2023b) under 50% and 60% compression ratios are presented in Table 12. A similar trend can also be observed here: incorporating both REC and PLC leads to a substantial reduction in perplexity across all settings. Moreover, ERC-SVD continues to outperform all existing baselines, further validating the effectiveness of these two components under these compression ratios.

## A.4 COMPARISON WITH PRUNING

Table 14 shows the performance of LLaMA-2-7B compressed by LLM-Pruner (Ma et al., 2023) and ERC-SVD under different compression ratios on the WikiText-2 dataset. It can be observed that ERC-SVD consistently outperforms the pruning method, achieving a perplexity of 58.88 under 60% compression ratio, compared to 114.23 for LLM-Pruner.

## A.5 DEMONSTRATION OF GENERATED CONTENTS

Table 15 shows the generation contents of models compressed by ERC-SVD compared to the original model under zero-shot conditions. The results indicate that across various input questions, compressed models consistently produce fluent, coherent, and highly relevant responses. Even at 40% compression ratio, the compressed model can also deliver accurate and relevant answers to common sense questions. These results highlight the effectiveness of ERC-SVD in achieving substantial compression without compromising performance.

Table 11: Zero-shot accuracy results across larger-scale LLMs under 20% compression ratio.

| MODEL | METHOD | Openb.↑ | ARC_e↑ | WinoG.↑ | ARC_c↑ |
|---|---|---|---|---|---|
| LLaMA-13B | SVD-LLM | 0.27 | 0.62 | 0.66 | 0.33 |
| | **ERC-SVD** | **0.28** | **0.69** | **0.69** | **0.39** |
| LLaMA-30B | SVD-LLM | 0.29 | 0.72 | 0.73 | 0.38 |
| | **ERC-SVD** | **0.30** | **0.73** | **0.73** | **0.43** |
| LLaMA-2-13B | SVD-LLM | 0.27 | 0.63 | 0.66 | 0.31 |
| | **ERC-SVD** | **0.31** | **0.67** | **0.68** | **0.38** |
| OPT-13B | SVD-LLM | 0.24 | 0.62 | 0.64 | 0.30 |
| | **ERC-SVD** | **0.24** | **0.63** | **0.65** | **0.30** |
| OPT-30B | SVD-LLM | **0.29** | 0.67 | 0.65 | 0.33 |
| | **ERC-SVD** | 0.28 | **0.68** | **0.67** | **0.34** |

Table 12: Ablation results of REC and PLC on LLaMA-2-7B (Touvron et al., 2023b) under 50% and 60% compression ratios.

| RATIO | METHOD | REC | PLC | C4↓ | Openb.↑ | ARC_e↑ | WinoG.↑ | HellaS.↑ | ARC_c↑ | PIQA↑ | MathQA↑ | Average↑ |
|---|---|---|---|---|---|---|---|---|---|---|---|---|
| 50% | ASVD | - | - | NaN | 0.13 | 0.26 | 0.50 | 0.25 | 0.23 | 0.50 | 0.20 | 0.30 |
| | SVD-LLM | - | - | 129.71 | 0.14 | 0.30 | 0.50 | 0.28 | 0.20 | 0.54 | 0.23 | 0.31 |
| | **ERC-SVD** | ✓ | ✗ | 126.61 | 0.14 | 0.30 | 0.53 | 0.29 | 0.21 | 0.55 | 0.24 | 0.32 |
| | | ✗ | ✓ | 117.79 | 0.13 | 0.33 | 0.54 | 0.29 | 0.22 | 0.58 | 0.22 | 0.33 |
| | | ✓ | ✓ | **100.34** | 0.14 | 0.35 | 0.55 | 0.31 | 0.22 | 0.59 | 0.22 | **0.34** |
| 60% | ASVD | - | - | NaN | 0.15 | 0.25 | 0.50 | 0.25 | 0.23 | 0.52 | 0.12 | 0.29 |
| | SVD-LLM | - | - | 263.02 | 0.14 | 0.26 | 0.50 | 0.27 | 0.20 | 0.53 | 0.21 | 0.30 |
| | **ERC-SVD** | ✓ | ✗ | 256.38 | 0.13 | 0.26 | 0.49 | 0.27 | 0.20 | 0.53 | 0.23 | 0.30 |
| | | ✗ | ✓ | 260.01 | 0.14 | 0.29 | 0.50 | 0.26 | 0.18 | 0.53 | 0.21 | 0.30 |
| | | ✓ | ✓ | **255.70** | 0.13 | 0.29 | 0.52 | 0.28 | 0.21 | 0.55 | 0.22 | **0.31** |

## A.6 COMPUTATION COMPLEXITY

ERC-SVD decomposes the original weight matrix $\boldsymbol{W} \in \mathbb{R}^{m \times n}$ into two low-rank matrices: $\hat{\boldsymbol{U}}_r \in \mathbb{R}^{m \times r}$ and $\hat{\boldsymbol{V}}_r \in \mathbb{R}^{r \times n}$. The layer compression ratio $R_l$ is computed as $R_l = 1 - \frac{(m+n)r}{mn}$. Under a fixed overall compression ratio $R_o$, the relationship between $R_l$ and $R_o$ is given by $R_l = \frac{NR_o}{k}$, where $N$ is the total number of layers and $k$ denotes the number of last layers to be compressed.

Given an input activation $\boldsymbol{X} \in \mathbb{R}^{n \times m}$, the original output is computed as $\boldsymbol{Y} = \boldsymbol{W}\boldsymbol{X}$. In the compressed model layer, an intermediate state is first computed as $\boldsymbol{I} = \hat{\boldsymbol{V}}_r\boldsymbol{X}$, followed by $\boldsymbol{Y} = \hat{\boldsymbol{U}}_r\boldsymbol{I}$. The computational complexity of the original model is $N \cdot \mathcal{O}(m^2n)$. For the compressed model, the first $(N - k)$ layers remain uncompressed and retain a complexity of $(N - k) \cdot \mathcal{O}(m^2n)$, while the compressed $k$ layers incur a cost of $k \cdot \mathcal{O}(m^2r + rnm)$. Thus, the total complexity becomes:

$$(N - k) \cdot \mathcal{O}(m^2n) + k \cdot \mathcal{O}(m^2r + rnm). \tag{10}$$

And the rank $r$ is given by:

$$r = \frac{mn(1 - R_l)}{m + n} = \frac{mn(k - NR_o)}{k(m + n)}. \tag{11}$$

Substituting the expression for $r$ into Equation 10, we obtain the simplified total complexity:

$$\boxed{N(1 - R_o) \cdot \mathcal{O}(m^2n)} \tag{12}$$

Compared to the original computation complexity $N \cdot \mathcal{O}(m^2n)$. This indicates that the total computation cost is reduced proportionally to the overall compression ratio. For example, if $R_o = 40\%$, the compressed model requires only 60% of the original computational cost.

Table 13: Zero-shot accuracy (↑) for seven common sense reasoning datasets on OPT-6.7B, Mistral-7B, and Vicuna-7B under 30% compression ratio.

| MODEL | METHOD | Openb.↑ | ARC_e↑ | WinoG.↑ | HellaS.↑ | ARC_c↑ | PIQA↑ | MathQA↑ | Average↑ |
|---|---|---|---|---|---|---|---|---|---|
| OPT-6.7B | SVD-LLM | **0.27** | 0.40 | 0.49 | 0.33 | 0.22 | 0.50 | 0.20 | 0.34 |
| | **ERC-SVD** | 0.22 | **0.56** | **0.61** | **0.40** | **0.25** | **0.69** | **0.24** | **0.42** |
| Mistral-7B | SVD-LLM | 0.13 | 0.44 | 0.53 | 0.30 | 0.20 | 0.62 | 0.20 | 0.34 |
| | **ERC-SVD** | **0.14** | **0.46** | **0.58** | **0.33** | **0.22** | **0.63** | **0.24** | **0.37** |
| Vicuna-7B | SVD-LLM | 0.22 | 0.50 | 0.56 | 0.35 | 0.26 | 0.63 | 0.21 | 0.39 |
| | **ERC-SVD** | **0.23** | **0.51** | **0.60** | **0.39** | **0.30** | **0.67** | **0.22** | **0.42** |

Table 14: Results on Wikitext-2 under different compression ratios ("RATIO") of LLaMA-2-7B.

| RATIO | LLM-Pruner | **ERC-SVD** |
|---|---|---|
| 20% | 8.09 | 7.63 |
| 30% | 12.59 | 10.32 |
| 40% | 20.36 | 14.17 |
| 50% | 40.97 | 24.26 |
| 60% | 114.23 | 58.88 |

## A.7 ERC-SVD ON VISION-LANGUAGE MODEL

In this section, we apply ERC-SVD to the language part of vision-language models (VLMs) such as LLaVA (Liu et al., 2024b), and evaluate its performance on image captioning (Section A.7.1) and visual question answering (Section A.7.2) tasks. All images used in these evaluations are selected from the ImageNet (Deng et al., 2009) dataset.

### A.7.1 PERFORMANCE OF IMAGE CAPTIONING

As shown in Figure 8, LLaVA-1.5-7B compressed with ERC-SVD under 20% compression ratio is still able to produce image captions that are faithful to the visual content. This indicates that the model retains its ability to interpret and describe images accurately, despite parameter reduction.

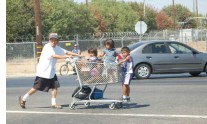
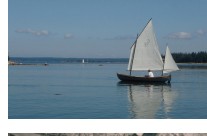

The image features a man pushing a cart filled with children, with four children sitting in the cart. He is walking down a street, with cars nearby. One car can be seen on the right side, and another car is further away on the right. Another car is located further behind the man, near the left edge of the image. The man appears to be taking care of his children by pushing the cart through the street.

The image features a sailboat sailing through a body of water, possibly a lake. A person is sitting in the boat, steering it with the sail. The boat's sails are white, indicating it's a sailboat. The boat's presence in the water suggests that it's enjoying a calm day, with no storms or rough waters. In addition to the boat, there are several small boats floating in the water.

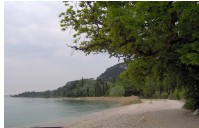
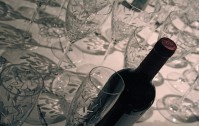

The image features a beautiful beach with sand and trees surrounding it. A tree stands near the beach and appears to be growing near the water. The tree's branches extend out over the beach, adding to the scene's scenic charm. A pathway leads to the beach, providing access to the sandy shore. There are also some trees along the beach, adding more visual interest to the scene.

In this image, there is a wine bottle sitting on a table next to a number of wine glasses. The wine bottle appears to be red, while the glasses are white in color. The glasses are arranged around the bottle, with some positioned closer to the bottle and others further away. These glasses seem to be ready for wine to be poured into them, creating a scene for wine consumption.

Figure 8: Image captioning results of LLaVA-1.5-7B compressed by ERC-SVD under 20% compression ratio. Captions that accurately describe the image content are highlighted in orange.

### A.7.2 PERFORMANCE OF VISUAL QUESTION ANSWERING

Figure 9 presents the visual question answering results of LLaVA-1.5-7B compressed by ERC-SVD. The model retains strong performance across these question types, indicating that its multi-modal reasoning capability remains intact despite the compression.

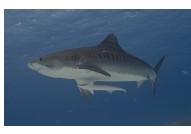
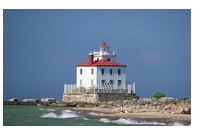

Q: *What should I do if I meet the shark in the ocean?*
A: When encountering a shark in the ocean, swimming away from it and avoiding close contact is crucial. Sharks are known to be aggressive predators, and their presence can pose a threat to humans. Swimming away from the shark while maintaining distance and swimming at a faster speed can help avoid being targeted. Additionally, ... wearing a safety cage or wearing a protective suit can be helpful. Finally ...

Q: *What are the things should be cautious while visiting?*
A: When visiting the lighthouse located on a rocky beach, one should be cautious about the rocky terrain and the rough sea conditions. The rocky beach might be unstable and difficult to navigate, while the rough sea may pose risks to swimming or boating. Additionally, the lighthouse's location atop the rocks may pose potential hazards if the lighthouse's structure is fragile or weak. To avoid potential accidents ...

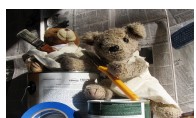
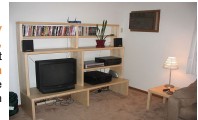

Q: *What is interesting about this image?*
A: The image is interesting because it features two teddy bears dressed in old clothes and painted to look like men, each wearing paint brushes. The teddy bears are placed next to a paint can, which suggests that they might be part of a painting project. This unusual scene creates a unique combination of objects that deviates from the typical depiction of teddy bears and paint brushes, adding a sense of creativity.

Q: *What do you think I can do in this area?*
A: In this area, you can watch TV, listen to music, and possibly listen to the radio. The TV is located on a wooden shelf, which also houses other items such as books, and there is a potted plant next to it. A small wooden table nearby offers a place to place other items. Additionally, there is a light in the area that illuminates the room, making it suitable for sitting and watching TV or listening to music. The presence ...

Figure 9: Visual question answering outputs generated by LLaVA-1.5-7B compressed using ERC-SVD under 20% compression ratio. Questions (Q) and model answers (A) are provided, correct answers are highlighted in orange to emphasize answer quality retention.

Table 15: Generated contents of LLaMA-7B (Touvron et al., 2023a) compressed by ERC-SVD.

| QUESTION | COMP. RATIO | ANSWER |
|---|---|---|
| What is the responsibility of the AI assistant? | - | The AI assistant is responsible for understanding the context and taking appropriate action. The AI assistant is responsible for understanding the intent, context, environment and taking appropriate actions. The AI assistant is responsible for interpreting. |
| | 20% | The AI assistant should be responsible for providing an answer to questions. This means that when asked a question, you should do your best to answer it and not give any information other than what is requested. You should also consider the tone and intent. |
| | 30% | The AI Assistant's job is to provide assistance to those who wish for assistance in their lives. It has no duty or obligation to assist those who do not desire assistance, nor does it have any duty to prevent. |
| | 40% | The AI assistant does not have a specific role. The AI assistant will provide instructions to follow a path, and will give directions as an instruction to follow the path. |
| What is the universe? Explain this concept. | - | The Universe is everything that exists. It includes all space, time, matter and energy. It also includes everything which has happened or will happen. The universe began about 13.7 billion years ago (the Big Bang) and will. |
| | 20% | The universe is the whole of space and time. It consists of all things in space and all things that happen, or have happened. There are many theories on how it began; there were theories that it had no beginning at all. |
| | 30% | The universe consists of everything in it, including matter and objects, space, the cosmos or outer space, and the galaxy system, as well as other galaxies that make up the Universe. The universe has its own life cycle. |
| | 40% | In the universe, life is very diverse and complex. The universe is made up of many different objects. These objects are divided into groups. Each group consists of different objects that have specific properties. This group contains the universe, which is large. |
| Why do cars need fuel to run? | - | Fuel is the lifeblood of a car. Without it, there's nothing to power the pistons and no way to drive. A car needs a constant supply of fuel. |
| | 20% | The fuel in the tank is burned to produce heat, which produces steam that drives a turbine, which turns a crankshaft. The crankshaft turns the wheels of the car, propelling it forward. |
| | 30% | The fuel is the fuel in a car and it burns and goes up and up. It's important because it burns the gasoline, which makes it go faster. It's important because you can get energy out. |
| | 40% | The fuel gives energy for the cars. The gasoline is used by a motor, which produces heat and air, heating up the engine. This allows the engines to travel faster, allowing them to reach higher speed. |
| Why do people feel hungry after exercise? | - | The answer is because the body burns more calories when you are exercising. You may not realize that the body uses up energy to digest food and absorb nutrients from it. |
| | 20% | Because the body is more efficient at converting food into energy when inactive, so when active it needs to convert food into energy faster. This is why people often get hungry after exercise. |
| | 30% | The reason people feel hungry after an activity is because they burn energy and energy is metabolised in the body. |
| | 40% | The reason the body is hungry after a long athletic activity was discovered by scientists in 1976. |

## A.8 STATEMENT OF LLM USAGE

We utilized large language models to assist in refining the writing of this paper, including improvements in grammar and clarity.