# OpenReview forum: "ERC-SVD: Error-Controlled SVD for Large Language Model Compression"
_ICLR.cc/2026/Conference — ICLR 2026 Conference Withdrawn Submission_

### Official Review · Reviewer_rhMX · 2025-10-30

**Soundness:** 2
**Presentation:** 3
**Contribution:** 2
**Rating:** 2
**Confidence:** 4

**Summary:**

ERC-SVD proposes splitting a fixed target rank into an intermediate rank (SVD on the original weight matrix) plus a small residual rank (SVD on the residual) and combining them, together with selecting only the last K layers for compression. The method is simple and empirically often improves on several baselines across many setups, but the paper falls short on novelty, theoretical justification, arbitrary experimental setting — therefore I recommend rejection.

**Strengths:**

Good experiment results

**Weaknesses:**

1. Limited novelty. The core contribution is simply applying SVD twice—first to the weight matrix, then to the residual. This is not a novel algorithmic insight.
2. Lack of  theoretical justification. The authors didn’t explain why two-stage SVD should outperform single-stage decomposition given the same rank budget. There's no analysis of what properties of the residual matrix that make separate decomposition beneficial, no approximation error bounds, and no indicators of when the method works or fails. The paper merely shows "we tried it, it works" without deeper understanding.  Without such theoretical justification,  I am concerned about the experimental results, as it has been proved long ago that Top-r SVD is already optimal, unless their loss is not the standard Frobenius norm or the whitening process changes the effective metric. It is possible that the LLM weight matrix has some special features, as many previous work pointed out the mismatch of the performance and the approximation error.
3. The design choices appear arbitrary. The residual compensation factor β is fixed at 0.05 across all layers, and compression ratios without justification. The limited sensitivity analysis tests only a narrow range and claims "robustness," but this actually may indicate the lack of principled design.

**Questions:**

Have you tested compressing the first k layers, middle k layers, or random k layers? What happens if you compress every other layer?

---

### Official Review · Reviewer_7ZH4 · 2025-10-30

**Soundness:** 3
**Presentation:** 3
**Contribution:** 3
**Rating:** 4
**Confidence:** 4

**Summary:**

This paper proposes **ERC-SVD**, a post-training SVD-based compression method for large language models (LLMs).
It introduces two components:


1. **Residual Compensation (REC)** – applies an additional SVD to the residual matrix left after truncation to reduce information loss.
2. **Partial-Layer Compression (PLC)** – compresses only the last *k* layers under a fixed overall compression ratio to alleviate forward error accumulation.


Experiments on a wide set of models (LLaMA, OPT, Mistral, Vicuna, Qwen) and tasks (language modeling, zero-shot reasoning, and vision-language benchmarks) show consistent improvements over ASVD, SVD-LLM, and AdaSVD, with stable behavior under different compression ratios.

**Strengths:**

1. Clear design; both REC and PLC contribute measurable improvements.
2. Compatible with quantization (GPTQ) and generalizes to VLMs (LLaVA).

**Weaknesses:**

1. **Limited theoretical depth**:

   - Eq. (5) defines the *effective scale*: $\alpha = \frac{m n}{m + n}$.
   But no derivation or intuition is given for why this specific harmonic-like mean is appropriate for rank scaling.

   - The per-layer rank formula: $r = (1 - R_\ell)\,\alpha$.
   This is heuristic and not theoretically motivated; it is unclear why the compression ratio $R_\ell$ interacts linearly with $\alpha$.

   - The residual compensation rank: $r_r = \alpha\,\beta$ (with $\beta = 0.05$), but without justification for this constant or explanation of its influence on stability and reconstruction accuracy. In particular, it is unclear **why $r_r$ should scale linearly with the matrix “effective size” $\alpha$** rather than being derived from empirical reconstruction error or singular value decay. A theoretical or empirical rationale for this proportionality assumption would substantially strengthen the paper.


2. If my understanding is correct, the model’s final performance should be primarily determined by the discrepancy between the **compressed and original outputs of the final layer**.
In this case, concentrating the compression budget on the **first few layers** would indeed cause larger propagated errors, but distributing the same compression ratio **uniformly or toward the middle layers** (e.g., layers 2–32) should not necessarily lead to an error explosion in the last layer. Therefore, it is unclear **why the authors always start compressing from the last layers**, instead of allocating layer-wise compression ratios according to the *incremental reconstruction error* or sensitivity of each layer. Moreover, the authors should provide a **clear table of per-layer compression ratios and resulting ranks** under different overall compression targets (e.g., 20%, 40%, 60%), to make the method reproducible and to verify whether the layer-wise allocation aligns with the claimed error-control principle.


3. **Missing comparison with latest and conceptually closest baselines**
 The paper only compares with ASVD, SVD-LLM (2025), and AdaSVD, but omits several important recent works that are directly relevant:
   - SVD-LLM v2(https://arxiv.org/abs/2503.12340)
   - Basis Sharing(https://openreview.net/pdf?id=gp32jvUquq)
   - Dobi-SVD(https://openreview.net/pdf?id=kws76i5XB8)


4. **Incomplete experimental reporting and outdated baselines**
 In Table 10, the authors only report results at a single compression ratio (20%) and compare solely against SVD-LLM.
 For a fair and convincing evaluation, results at **all tested compression ratios (20%, 30%, 40%, 50%, 60%)** should be reported, together with comparisons to other strong and recent baselines, including **SVD-LLM v2**, **Basis Sharing**, and **Dobi-SVD**.
 Moreover, most experiments should focus on newer architectures like **LLaMA-3**, **Qwen-2.5**, **Qwen-3**, not older models such as LLaMA-7B or LLaMA-2-7B.

**Questions:**

Please see the weaknesses section above.

---

### Official Review · Reviewer_3F7n · 2025-10-31

**Soundness:** 3
**Presentation:** 3
**Contribution:** 3
**Rating:** 2
**Confidence:** 4

**Summary:**

This paper proposes two main techniques to improve SVD-based low-rank approximation for neural network compression: (1) residual compensation for SVD truncation, which reportedly outperforms traditional SVD-based methods; and (2) partial-layer compression, which focuses the SVD truncation on layers closer to the LM head to reduce output error. The paper also conducts extensive experiments to validate these ideas, demonstrating improvements over existing approaches.

**Strengths:**

1.The idea of partial-layer compression for SVD is novel and addresses a meaningful issue in SVD-based compression.

2.Experimental design is thorough, with comprehensive coverage of relevant evaluation points.

**Weaknesses:**

The paper demonstrates a lack of mathematical understanding regarding SVD: from a theoretical perspective, the proposed residual compensation for SVD truncation is equivalent to a single SVD truncation and does not provide additional benefit.

The choice of partial compression ratio lacks theoretical justification or empirical explanation. Recent works such as SHEARED LLAMA use a learn-then-prune strategy for structured pruning, which is not considered here. The proposed “error” metric is also not used as a basis for further method selection or ablation studies.

Results for high compression ratios are not compared against smaller pretrained models with similar compute, leaving open the question of relative effectiveness.

here are typos in the paper: for example, many figures still use “resSVD” even though the method is named ERC-SVD.

The baseline for SVD-LLM is incomplete: it should include Alpaca-based parameter retraining, but the paper appears to only implement the first stage, resulting in weaker baseline performance. For instance, the LLaMA-7B replication underperforms compared to reported SVD-LLM results.

Other SVD baselines are also applied as whole-network compression, which may be an unfair comparison. Combining partial-layer compression with other SVD variants could further validate the effectiveness of the proposed methods.

**Questions:**

1.Did you consider whether the gain from residual compensation for SVD truncation is due to calculation precision or stochastic variation? Were any ablation experiments performed to isolate this effect?

2.Can you clarify exactly which layers are compressed at each partial compression ratio? The paper lacks details about per-layer compression ratios, making it difficult to manually verify computational fairness. Is there any risk of unintentional unfairness due to how compression ratios are defined (since SVD truncation at rank r requires storing two matrices, so the compression rate is actually r(n+m)/nm)?

3.Is the reported speedup consistent with practical LLM scenarios? In real-world deployments, LLMs often scale with output length, increasing the proportion of attention computation. Thus, the speedup may not be as significant as reported for seq_len=32.

---

### Official Review · Reviewer_B5yp · 2025-11-01

**Soundness:** 2
**Presentation:** 2
**Contribution:** 2
**Rating:** 2
**Confidence:** 4

**Summary:**

This paper proposes ERC-SVD, a post-training SVD-based LLM compression method, to address the issues of large truncation loss and severe error propagation in existing SVD solutions. Its core innovations are: 1) Two-stage residual compensation, utilizing the residual matrix generated by SVD truncation to reduce loss; 2) Partial-layer compression, compressing only the last few layers of the model under a fixed overall compression ratio to minimize error accumulation. Validated on 12 models from 5 LLM families and 10 benchmark datasets, ERC-SVD consistently outperforms baselines like traditional SVD and ASVD. For instance, under 30% compression ratio, the PTB perplexity of LLaMA-2-7B is reduced by 75% compared to SVD-LLM. It can also be extended to VLMs and combined with quantization.

**Strengths:**

- Two-stage residual compensation and partial-layer compression provide a new perspective for SVD compression.
- Rigorous theory and strict experimental control ensure reliable results.
- Complex technologies are described in plain language, with formulas and pseudocode for high readability.

**Weaknesses:**

- Only the calculation process for residual compensation is given, without more detailed theoretical derivation from the loss function to explain why this method is effective.
- A notable limitation of the experiments presented in the paper lies in the selection of baselines for comparison—specifically, the exclusion of newer SVD-based LLM compression methods that are highly relevant to the research topic, such as Dobi-SVD.

**Questions:**

- The theoretical derivation needs to be supplemented to explain why residual compensation can reduce errors.
- There was no comparison with newer baselines, such as Dobi-SVD.
- The largest tested model (7B parameters) falls short of demonstrating scalability to larger-scale models (e.g., 70B), which are more representative of real-world deployment scenarios.

---

### Note · Authors · 2025-11-12

I have read and agree with the venue's withdrawal policy on behalf of myself and my co-authors.